# Gross Anatomy of the Female Reproductive System of Sugar Gliders (*Petaurus breviceps*)

**DOI:** 10.3390/ani13142377

**Published:** 2023-07-21

**Authors:** María del Mar Yllera, Diana Alonso-Peñarando, Matilde Lombardero

**Affiliations:** 1Unit of Veterinary Anatomy and Embryology, Department of Anatomy, Animal Production and Clinical Veterinary Sciences, Faculty of Veterinary Sciences, University of Santiago de Compostela–Campus of Lugo, 27002 Lugo, Spain; mar.yllera@usc.es; 2Department of Animal Pathology, Faculty of Veterinary Sciences, University of Santiago de Compostela–Campus of Lugo, 27002 Lugo, Spain; diana.alonso.penarando@rai.usc.es; 3DVM at Veterinary Clinic Madivet, Calle Comercio, 5, Bargas, 45593 Toledo, Spain

**Keywords:** sugar glider, female genital anatomy, ovary, uterine tuba, uterus, vaginal complex, clitoris

## Abstract

**Simple Summary:**

Sugar gliders are small marsupials that are becoming quite common as pets. As patients, they are challenging for clinical veterinarians. References about exotic animals are difficult to find, especially regarding the internal anatomy of *Petaurus breviceps*, knowledge of which is essential to reach a correct diagnosis of pathologies. Since the reproduction of marsupials is peculiar, we focused on the female genital apparatus of this species, very different from that of the usual mammals in veterinary practices. Sugar gliders have duplicates of all the genital organs: they have two ovaries, two oviducts, two uteri and two vaginas. In addition, in pregnant females, a new conduit arises: the birth canal, through which the offspring will leave the genital tract to go outside. In this species, the urinary, genital and digestive systems flow into a small cavity called the cloaca, which receives urine, the foetus and faeces, and communicates with the outside through a single orifice. Thus, the female genital apparatus of the sugar glider fits the general schema of marsupials.

**Abstract:**

We dissected carcasses of eight mature females, both parous and non-parous specimens, to study the macroscopic anatomy of the female reproductive system in the sugar glider. The genital system includes double organs, namely the right and left ones, which are completely separated. It includes two ovaries, two oviducts, two *uteri* and a vaginal complex. The *uteri* are fusiform-shaped and lack horns. The vaginal complex includes two lateral *vaginae* and a median vagina, also called the ‘birth canal’. The cranial end of both lateral *vaginae* partially fuses, forming an expansion named the vaginal sinus, which is divided into two parts by a longitudinal septum, one for each vagina, where the ipsilateral uterine cervix opens. The caudal end of the lateral *vaginae* opens into a medial and impar duct: the urogenital sinus that serves as a common passage for the reproductive and urinary systems. In non-pregnant females, only the lateral *vaginae* are present. In pregnant and recently parous females, a short median vagina extends from the caudal wall of the vaginal sinus to the cranial end of the urogenital sinus. In the ventral wall of this sinus, next to its caudal opening, there is a forked clitoris.

## 1. Introduction

Exotic animal clinical practice is challenging for veterinarians because of the widely varying anatomy, peculiar physiology and specific needs of the new companion animals. Very often, it is hard to find appropriate references in the literature, and even sometimes, the available information is contradictory, and erroneous extrapolations from domestic species are applied. Knowledge of anatomy is basic to reach a correct diagnosis of pathologies and to interpret the images obtained with modern diagnostic imaging methods. The purpose of this article is to provide a practical approach to the female genital anatomy of the sugar glider.

Sugar gliders (*Petaurus breviceps*) are becoming quite popular as pets. They are originally from Australia and New Guinea, but were introduced into Tasmania. They arrived as pets to the USA in the 1990s [1] from Western New Guinea (Indonesia) [2], and much later to Europe. They are small, nocturnal and omnivorous marsupials that live in arboreal habitats and have a large gliding membrane named the *patagium* between the front and hind limbs. Therefore, they can glide long distances, up to 50 metres, between trees [3]. Sugar gliders are a social species that builds communitarian nests in holes in eucalypts. In the wild, they feed on plants, insects and arthropods’ exudates, with a natural diet depending on the season and habitat [4,5]. Furthermore, some Tasmanian populations have carnivorous preferences, as they feed on eggs and young birds [6].

The body organization of the three great groups of mammals, eutherians, and marsupials and monotremes, is very similar, but they differ in their reproductive performance, which leads to differences in the morphology of their genital apparatus. In marsupials, the urinary, genital and digestive systems end in a common space, the cloaca, which communicates with the exterior. The newborns, named joeys, are in an early stage of development and after a short period of gestation (15–17 days in the sugar glider) [7,8], a highly altricial young emerges through the cloacal opening and climbs by its own into a pouch called the marsupium, where its development will be completed.

Although there is abundant literature on marsupials, there are practically no articles on the anatomy of *Petaurus breviceps*, especially regarding its internal organs. The purpose of this study is to contribute to a better understanding of the female genital apparatus of this species.

## 2. Materials and Methods

Eight females were investigated, all of whom were adults and intact specimens. They died from different pathologies, and their carcasses were donated by their owners to carry out this study. Seven animals had given birth to joeys at least once, including three females with joeys in the pouch, one with two joeys recently weaned, and three individuals that had not been pregnant for some years; one additional female was non-parous.

Seven specimens were kept frozen and one was preserved in alcohol 70° until the time of the study. The dissections were performed by accessing the body cavities through the ventral midline. In two cases, after the macroscopic study, the entire urogenital apparatus except the kidneys was removed and preserved in formalin for a second and more detailed study.

All the dissections were carried out under a Nikon AFX-II magnifying device (magnification from 0.66 up to 4.0, with oculars of 10×), and photographs were taken with a Nikon Coolpix S6 camera (Zoom Nikkor ED 5.8–1.7 mm 1:3.0–5.4) equipped with a macro lens.

## 3. Results

Female sugar gliders have a pouch in the ventral part of the abdomen. Its opening is placed in the midline, caudal to the *umbilicus*. The pouch follows the longitudinal axis of the animal; it has two lateral pockets and contains four *mamae* (nipples or teats): two in each pocket, one cranial and one caudal. The mucosa that lines the inner wall of the pouch has secretory glands, which are macroscopically visible (Figure 1).

The female genitalia are in the caudal half of the abdominal cavity, and they are related dorsally to the rectum and ventrally to the urethra and the urinary bladder. The sugar glider female reproductive organs include paired ovaries, paired uterine tubes, paired uteri and a vaginal complex that includes paired lateral *vaginae* (Figure 2 and Figure 3a).

The ovaries are close and dorsolateral to their ipsilateral uterus, near the junction between the uterus and the uterine tube. They have a brownish colour and ellipsoidal form and are flattened dorsoventrally. In this study, all the females were pubescent, so the gonad’s surface was irregular due to the projection of follicles and/or corpora lutea. Thus, all functional ovarian structures are visible. Each ovary is attached to the abdominal roof by a long ligament known as the *mesovarium*. The cranial free border of the *mesovarium* is a strong band called the suspensory ligament of the ovary (*ligamentum suspensorium ovarii*) (Figure 4a). The *mesovarium* also supports the proper ligament of the ovary (*ligamentum ovarii propium*), a band of connective tissue covered with peritoneum, which extends from the caudal pole of the gonad (*extremitas uterina ovarii*) to the adjacent tip of the uterus. As the peritoneal folds do not cover the dorsal surface of the ovary, an *ovarian bursa* cannot be described in this species (Figure 4b).

Uterine tubes or oviducts (*tubae uterinae*) are long and flexuous. They begin next to the corresponding ovary and lead to the uterus on the same side. The free cranial extremity of the tube is funnel-shaped and has a very thin wall. It is called the infundibulum (*infundibulum tubae uterinae*) and is close to the ipsilateral ovary. The rest of the oviduct is tubular and very convoluted. It is divided into two parts: the proximal and narrower one, usually called the *ampulla* (*ampulla tubae uterinae*), and the most coiled part, which is the *isthmus* (*isthmus tubae uterinae*). Each tuba is supported by a serous fold called the *mesosalpinx*, a lateral fold released from the *mesovarium*. The *mesosalpinx* and *mesovarium* enclose a pouch that contains the tuba but not the ovary. This fact, together with the proximity of the ovary and the uterus, sometimes makes it difficult to visualize the tubes (Figure 4b).

The paired uteri are elongated and fusiform-shaped, with their major axis running obliquely to the corporal axis. Sugar glider uteri lack horns (*cornua uteri*), but they do have a body (*corpus uteri*) and a neck (*cervix uteri*). A ligament joins the medial surfaces of the caudal part of both uterine bodies. Although they are independent structures, both uteri appear to be fused caudally since they are surrounded by connective tissue (Figure 4). The caudal part of each uterus narrows and they run parallel to each other on each side of the median line forming the uterine neck or cervix, which projects onto a papilla into the vaginal complex. The cervix lumen is almost occluded by mucosal folds, almost hiding the orifice that communicates the interior of the vaginal complex with each uterus (*ostium uteri externum*) (Figure 2 and Figure 5).

The vaginal apparatus or vaginal complex includes two lateral *vaginae* and a median vagina, also called the ‘birth canal’. Lateral *vaginae* are long and curved in a U-shape. They are placed caudal to the uteri and form two loops located on either side of the midline. Each loop has two ends, one cranial and another caudal, very close to each other and located on the same plane and joined by a peritoneal fold. The cranial end of both lateral *vaginae* partially fuse, forming an expansion named the vaginal sinus (formerly called the ‘anterior vaginal expansion’ or ‘cul-de-sac’) (Figure 2, Figure 3b and Figure 5). It is divided by a longitudinal *septum* into two parts, one for each vagina, where the corresponding uterine cervix opens. This separation is complete in the non-parous specimen but incomplete in parous ones (Figure 5). The caudal end of the lateral *vaginae* opens into a medial and impar duct: the urogenital sinus that serves as a common passage for the reproductive and urinary systems (Figure 6). The median vagina, very short in the sugar glider, extends from the caudal wall of the vaginal sinus to the cranial end of the urogenital sinus in recently parous females but does not exist in non-pregnant animals (Figure 3c and Figure 7).

The urinary bladder is ventral to the median vagina and urogenital sinus (Figure 2a). A large external urethral orifice is located on the ventral wall of the urogenital sinus, the ending point of a short urethra. The ureters, originating in the ipsilateral kidney, run dorsolaterally to the uterus. Each ureter passes dorsally to the cranial branch of the ipsilateral lateral vagina and is incorporated into the peritoneum fold between the two branches. Their course is parallel to the caudal branch until they end in the vicinity of the neck of the urinary bladder, in a dorsal position. The two ureters terminate very close to each other by means of two elongated orifices located near to the internal urethral orifice, where they empty their content (Figure 5 and Figure 6).

The female sugar glider has neither a vulva nor an anus. Instead, there is a single opening, ventral to the beginning of the tail that communicates with a small internal cavity, the cloaca, a common organ of the digestive system and urogenital apparatus.

The urogenital sinus is a long tube that begins next to the caudal wall of the vaginal sinus and runs ventrally to the final part of the digestive system (large intestine) (*intestinum crassum*) to the vicinity of the cloacal orifice (Figure 7). It opens into the floor, close to the rectum that empties into the cranial part of the cloacal cavity.

The genital and urinary ducts empty into the cranial part of the urogenital sinus. The caudal loop of the lateral *vaginae* and the external urethral orifice (*ostium urethrae externum*) flow at the same level, but the lateral *vaginae* open into the dorsolateral wall, whereas the external urethral orifice is on the ventral one (Figure 6).

In the caudal part of the urogenital sinus, next to its cloacal opening, there is a ventral fossa (*fossa clitoridis*) where the clitoris is located. This pit is covered by two folds of the lateral mucosa, one right and one left, which practically hide the fossa. The clitoris is forked. Each of its halves is elongated, with a constant thickness except at its terminal end, near the cloaca, where it tapers to a conical point. Given its great length, at rest, its caudal part must be folded inside the fossa (Figure 8).

## 4. Discussion

The name ‘marsupials’ designates a group of mammals whose females have a pouch, or marsupium, where the young (joeys) finish their development after their birth. The marsupium is usually attached to the ventral wall of the abdomen, but there are considerable variations between species. Sometimes it opens cranially, sometimes caudally or centrally [9]. There are even marsupials that lack a pouch [9,10]. Russell (1982) [11] described six types of marsupium. In type 1, the mammary area is not covered by the skin although lateral ridges of skin develop during the breeding season; in type 2, the mammary area is partially covered; in types 3, 4, 5 and 6, the mammary area is covered by a fold of skin forming a real pouch; they differ in the position of their openings: central (type 3), cranial or anterior (types 4 and 5) and caudal (type 6). In addition, in type 4, the teats are located in two pockets, right and left, projecting forward from the cranial margin of the skin fold. Since the mammary area is completely covered by a skin fold and the opening of the bursa is located in its cranial part, the sugar glider marsupium is included in type 5, according to Russell’s (1982) [11] classification.

The morphology of the pouch in the sugar glider matches the description made by Smith in 1973 [7]. It opens near its cranial end and has two pockets extending onto the flanks of the female, where the *mamae* are. According to Tyndale-Biscoe and Renfree (1987) [12], Johnson-Delaney (2021) [13] and our own data, the female sugar glider has four *mamae* (nipples) inside the marsupium, two on the right and two on the left. Nevertheless, other authors point out that they are only two teats in the pouch [7,8,14,15]. Variations between individuals could be possible, as indicated by Smith (1973) [7] and Girling (2013) [16].

We observed macroscopic secretory glands in the mucosa lining the inner surface of the pouch. The presence of apocrine glands in the pouch skin has previously been described in the sugar glider by Smith (1973) [7] and in other marsupials such as the tammar wallaby [17] and red kangaroo [18]. Various functions have been suggested for the secretion of these glands. The role of the chemical protection of the young by the secretions are not well known, but they could have antimicrobial activity as found in the tammar wallaby, whose pouch skin expresses genes for the antimicrobial peptide cathelicidin [19]. The maternal glandular secretions could also provide moisture and their smell could play an important role in guiding the joey on its journey from the cloaca to the pouch after birth [18]. When the young marsupial is born, many physiological processes are still immature for a time after parturition, among them pulmonary respiration. The lungs are immature, so gas exchange takes place through the skin for some days. Inside the pouch, the humid environment may aid this integumental gas exchange [20].

Interestingly, although marsupials have a dual hormonal control of genital organ development, pouch development is not under this control [21]. The initial development of the pouch, as well as of the mammary glands and the scrotum in the male, is evident before the differentiation of the gonads and depends on the expression of a gene or genes on the X-chromosome [22]. In fact, it is the number of X-chromosomes that determines whether a marsupium or a scrotum will form: animals with one X-chromosome will develop a scrotum while those with two X-chromosomes will form a pouch [22,23].

The female genital apparatus of eutherian mammals basically consists of paired organs responsible for producing gametes (ovary) and a system of ducts that communicate with the outside. Each part of the genital tract is specialized to perform one or more functions. Thus, the ovary produces gametes and reproductive hormones, fertilization takes place in the oviduct (pair), and the uterus (unpaired, except in lagomorphs and most rodents which have duplex uteri) provides the environment and adequate nourishment for the embryo development until its birth. In eutherian mammals, the vagina (unpaired) serves both as copulatory organ and as birth canal. The vestibule continues the vagina to the vulva, which opens externally. The genital anatomy of the female sugar glider is similar to that of other marsupials and quite different from that of eutherian mammals. They have double reproductive organs: two ovaries, two oviducts, two uteri and two lateral *vaginae*. The origin of the double reproductive tract is the result of a difference between eutherian and marsupial mammals in the embryonic development of the genital and urinary ducts. In eutherians, when the ureters migrate from the roof of the celomic cavity ventrally to connect with the bladder, they are placed laterally to the paramesonephric (Müllerian) ducts. Therefore, these can fuse with each other to form the body of the uterus and the cranial portion of the vagina. Conversely, in marsupials the ureters migrate medially to the genital ducts, preventing their fusion [10].

As the ovaries were not enclosed in any bursa in the female sugar gliders that we studied, we suggest that there is no true ovarian bursa in this species unlike those described in some South American marsupials such as *Didelphis* sp. [24] and in *Dendrolagus* [25]. Johnson-Delany (2002) [14] described the ovaries as lying against the medioventral side of the uterus, near the junction of the uterus and the oviduct. However, in all the specimens we observed, the gonads were located dorsolateral to the uterus and could not be seen in a ventral view of the genitalia.

Uterine tubes vary in length and their degree of convolution depends on the marsupial species [26]. There are three parts of the mammalian uterine tubes: *infundibulum*, *ampulla* and *isthmus*. However, unlike in eutherian mammals, in the sugar glider, the widest tubular part is the *isthmus*, as seen in the anatomy of other marsupials referred to by Taggart (1994) [26] and Hughes (2000) [27].

Although the uterine function is similar in marsupial and eutherian mammals, there are important differences in uterine morphology. In marsupials, there are no uterine horns, only a body and neck (*cervix*) are present. The uterine neck length varies between species [26], as does the uterine body shape: usually, it is fusiform, as we have observed in the sugar glider, in accordance to that previously described by Johnson-Delany and Lennox (2017) [28], but was abnormally enlarged in the extinct Tasmanian tiger (*Thylacinus cynocephalus*) [27] and is flattened dorsoventrally in the Tasmanian devil (*Sarcophilus satanicus*) [29]. However, in most species, the caudal ends of both uteri (*cervices*) are surrounded by a common sheath of connective tissue [14,28], as we found in *Petaurus breviceps.*

The uterine papilla formed by the opening of each cervix into the vaginal sinus in the *Petaurus breviceps* was previously described by Johnson-Delany (2002) [14]. This author also reported an *os uteri* ventrolaterally on the papilla that, in contrast, we could not identify in our macroscopic study. This papilla seems to be quite common in marsupials since it has also been observed in *Sarcophilus satanicus* [29], in *Thylacinus cynecephalus* [27] and in *Dendrolagus* [25].

The function of the lateral *vaginae* is to facilitate the arrival of sperm to the *uteri* [30]. The male marsupial deposits semen in the upper part of the urogenital sinus, where the caudal ends of the lateral *vaginae* empty. From here, spermatozoa progress up through these ducts and penetrate the vaginal sinus, near the *cervices* [31,32]. Since there is no direct communication between the urogenital sinus and the *uteri*, this is the only possible pathway in all non-parous and most parous female marsupials. However, in macropods, the birth canal (median vagina) is permanent, so it allows direct access from the urogenital sinus to the cervices in parous females [33].

The shape of the lateral *vaginae* varies among marsupials. In *Petaurus breviceps*, they are long and U-shaped in accordance with our study, but they can become long and convoluted as in *Caenolestes obscurus* or short and straight such as in *Trichosurus vulpecula* and *Tarsupes rostratus*. Additionally, in some species, the cranial part of the lateral vagina expands to form a chamber that serves as a seminal receptacle [12].

In pregnant females, a median vagina is formed in the midline, lying between the lateral *vaginae* [7,16,22]. In most marsupials, this forms a transient birth canal in the medial connective tissue between the vaginal and urogenital sinuses. It opens before each birth and closes soon after the passage of the foetus, and reforms in each subsequent parturition [9]. However, in most *Macropodidae* (kangaroos), the median vagina becomes a permanent structure after the first birth [12]. Judah and Nutall (2008) [8] point out that it is a temporary structure in *Petaurus breviceps*, which coincides with our own macroscopical observations. However, the birth canal could persist even if it is blocked by tissue separating the vaginal and urogenital sinuses. Histological study would be required to confirm that it disappears completely after parturition, as seems to be the case in the Tasmanian tiger [27].

According to our studies, in the sugar glider, the medial vagina is very short whereas the lateral ones are long and thin. Thus, our results are in accordance with those previously described by Johnson-Delaney and Lennox (2017) [28]. However, it is not always so in marsupials. For instance, in opossums and macropods, the medial and lateral *vaginae* are nearly equal in length [28]. In *Petaurus breviceps*, both structures, the vaginal and the urogenital sinuses, are very close, but it could be the same as in other marsupials such as *phalangeridae*, *dasyuridae* or *vombatidae* [30]: they are contiguous, but there is no communication between them except in pregnant or recently calved females.

We observed an incomplete septum in the midline of the vaginal sinus, separating the beginning of the two cranial loops of the lateral *vaginae*, in parous females (six of seven specimens). This was previously described by Johnson-Delany (2002) [14] and Johnson-Delany and Lennox (2017) [28]. According to Rodger (2020) [32], in non-parous marsupial females this median vaginal septum separates the left and right parts of the genital tract, but after the first parturition, in most marsupials this septum is perforated and the right and left sides of the vaginal sinus communicate. Our observations support this statement in the sugar glider. In some species such as *Dendrolagus matschiei*, the division of the sinus vaginalis is only partial and due to a longitudinal fold, whose height decreases as it advances in the caudal direction until it disappears. It is only attached to the ventral wall but not to the dorsal one, only dividing the upper half of the sinus [25]. Differences in the septum between species (complete, incomplete, or absent) have previously been described by Tyndale-Biscoe and Renfree (1987) [12].

Because of the close relationship of the ureters and vagina in marsupials, unlike eutherian mammals, neutering of female sugar gliders is not a routine procedure [34]. Given the technical complexity of the surgery, the animal’s small size [34], and the fact that it has not been confirmed to have a favourable impact on their health [28], to avoid unwanted gestations in this species, only males are usually neutered [34]. However, due to the presence of the cloaca, which connects the urogenital and digestive tracts, this species is relatively prone to ascending infections from the cloaca, which can cause vaginitis and metritis [35]. Sometimes, these pathologies can be resolved conservatively [35]. However, when removal of the female genitalia (ovary-vagina-hysterectomy) is necessary for medical reasons, careful consideration should be given to the course of the ureters, isolating their last section from the ipsilateral lateral vagina. In obese animals it may not be possible to separate these ducts, in which case, only the ovaries and uteri could be removed (ovariohysterectomy) [14].

As stated by Pearson (1945) [30], the long urogenital sinus of the sugar glider seems to be characteristic of marsupials. He also described a clitoris in *Petaurus breviceps*, located on the ventral wall of the urogenital sinus, near its posterior extremity. According to Pavlicev et al. (2022) [36], there seems to be a relationship between the anatomy of the penis and that of the clitoris in marsupials: when the *glans penis* is bifurcated, the clitoris is also forked. Thus, a bifurcated clitoris was found in female short-tailed opossums [14], the Tasmanian devil (*Sarcophilus satanicus*) [29], the Tasmanian tiger (*Thylacinus cynecephalus*) [27] and the sugar glider (*Petarurus breviceps*). In contrast, it is unforked in macropodids [37]. Matthews (1947) [25] described the clitoris in *Dendrolagus ursinus* as a conical structure, flattened on its dorsal side, convex on its ventral side and lying in a pocket, so it did not project into the urogenital sinus lumen. In addition, Gonçalves et al. (2009) [24] reported the existence of erectile tissue in the ventral wall of the urogenital sinus, next to its end, forming a genital tubercle that resembles a clitoris.

The short urethra of the sugar glider follows the general rule of marsupials indicated by Pearson (1945) [30]. According to his studies, the only exceptions were the genders/genere *Parameles* and *Bettongia*, due to the cranial displacement of the urinary bladder. The urethral orifice (*ostium urethrae externum*), placed on the ventral floor of the urogenital sinus, was previously identified by Girling (2013) [16] at the same level of the lateral *vaginae* ends and had also been described in other species (Hughes, 2000) [27], although it does not seem to be a constant rule in marsupials. Thus, in some South American species, such as *Didelphis albiventris* [38], the external urethral orifice is located cranially at the end of the lateral *vaginae*, but caudally in *Dendrolagus ursinus* [25].

Although we did not observe it in the sugar glider, a papilla has been described at the end of the urethra (external urethral orifice) in some marsupial species [25].

The existence of a reduced cloaca, such as the sugar glider one, seems to be a common feature of marsupials [39], although in some South American species, such as *Didelphis albiventris* [38], *Chinorectes minimus* [40] and *Monodelphis* [36], the urogenital sinus and the anus open separately on the surface of the body.

## 5. Conclusions

Sugar gliders are lively, playful and intelligent marsupials that have gained recent popularity as exotic pets. They are a small, lovely and social species, and bond closely with their owners if given the socialisation they require. Like other pets, they are susceptible to various illnesses and they arrive at veterinary clinics. Although dystocias are not frequent because the joeys are very small when they are born, as mammals they can suffer from uterine inertia, genital neoplasias, malpositioned foetus, malformation of the reproductive tract and so on. Occasionally, it may be necessary to perform an ovariohysterectomy in cases of pyometra. On these and other occasions, clinicians need extensive knowledge of the female genital tract to reach a correct diagnosis when using imaging methods and performing surgery, if necessary. According to our study, the female genital system of the *Petaurus breviceps* has duplicate genital organs as it corresponds to a marsupial with a vaginal complex that opens into a long urogenital sinus, carrying both the foetus and urine to a cloaca, which also receives the faeces from the digestive system. Our results provide useful information for clinicians, especially for the correct interpretation of modern imaging techniques, such as magnetic resonance imaging (MRI) and computed tomography (CT), as well as to perform successful surgery in the female genital tract when needed.

## Figures and Tables

**Figure 1 animals-13-02377-f001:**
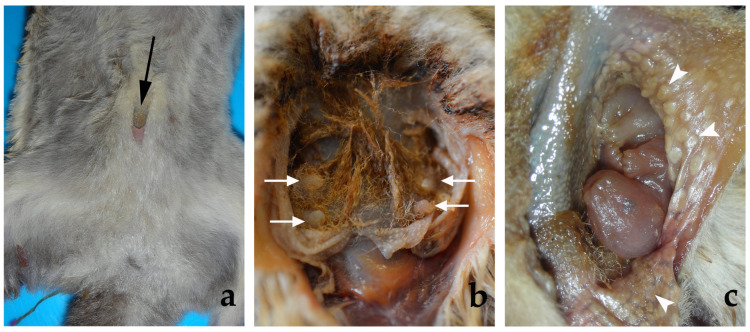
Ventral view of a female sugar glider’s abdominal wall. (**a**) The black arrow points to the opening of the pouch (*marsupium*). (**b**) *Marsupium*: the pouch was opened to show the four teats (*mamae*) (**⇨**). (**c**) *Marsupium*: the left part of the pouch was opened to show the secretory glands (
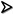
). A joey can be seen deep inside the pouch.

**Figure 2 animals-13-02377-f002:**
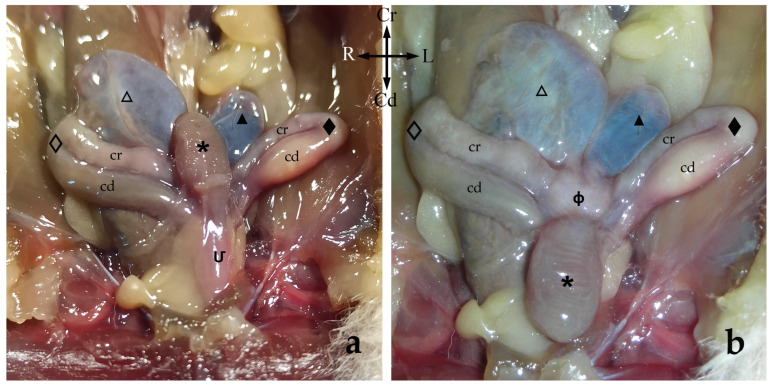
Ventral view of the female sugar glider’s apparatus urogenitalis in situ (**a**) and with the urinary bladder displaced caudally (**b**). Fresh specimen: (**Մ**) *sinus urogenitalis*; (**ф**) *sinus vaginalis*; (_🞳_) urinary bladder. Right-side structures are shown with empty patterns whereas left-side structures are marked with filled patterns: (**◇**, ◆) lateral *vaginae* with cranial loop (cr) and caudal loop (cd); (**△**, ▲) *uteri*.

**Figure 3 animals-13-02377-f003:**
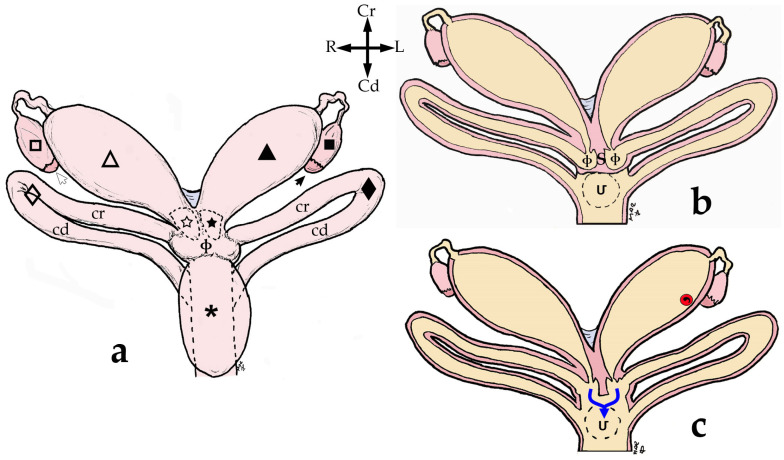
Schematic drawing of the genital system of the female glider: (**a**) intact; (**b**,**c**) opened to expose the lumen; the urogenital sinus is marked with a dotted line. (**a**) The (_🞳_) urinary bladder is reflected caudally. (**b**) Schema of the female genitals of a non-pregnant sugar glider, with two independent vaginal sinuses. (**c**) Schema of the female genitals of a pregnant sugar glider showing a single median vagina (confluent blue arrows). Right structures are represented by an empty pattern and left structures with a filled pattern: (**⇨**, **➞**) ovaries; (**□**, **■**) *infundibula* of the uterine tubes; (**△**, ▲) bodies of the *uteri*; (**☆**, ★) *cervices*; (**◇**, ◆) Lateral *vaginae* with cranial loop (cr) and caudal loop (cd); (**ф**) aginal sinus; (**Մ**) *Sinus urogenitalis*.

**Figure 4 animals-13-02377-f004:**
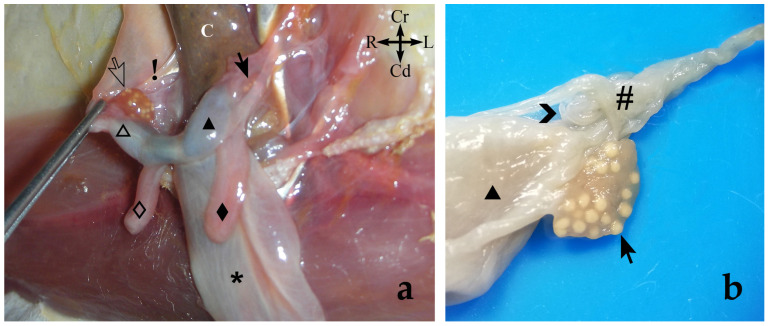
(**a**) Ventrolateral view of the sugar glider’s *apparatus urogenitalis* in situ. Fresh specimen: The urinary bladder is reflected caudally to expose the *uterii* and *vaginae*. The (**△**) right uterus has been displaced ventrally to display the right ovary and its peritoneal folds: (**⇨**) right ovary; (**!**) suspensory ligament of the ovary; (_🞳_) urinary bladder. Right structures are represented by an empty pattern and left structures with a filled pattern: (**◇**, ◆) lateral *vaginae*; (**△**, ▲) *uteri*; (**➞**) left ovary; (**C**) colon. (**b**) Dorsal view of the female sugar glider’s *apparatus genitalis* ex situ: ovary and uterine tube. Specimen is fixed in formalin. The oviduct is included in a pouch formed by the peritoneal folds (*mesovarium* and *mesosalpinx*); (▲) left uterus; (**>**) uterine tube; (**#**) pouch; (**➞**) left ovary.

**Figure 5 animals-13-02377-f005:**
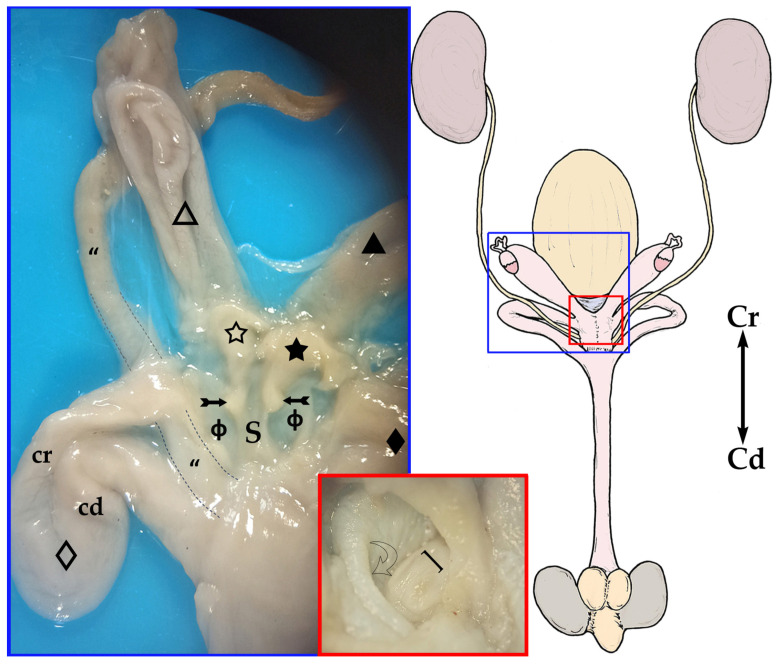
On the right, a general sketch of the female sugar glider urogenital system ex situ (dorsal view) with the areas’ detailed photographs. Blue inset: ventral view of the female sugar glider’s *apparatus genitalis* ex situ. The caudal parts of the uteri have been dissected to expose the cervix. The vaginal sinus was opened ventrally to show the uterine *papillae* and *septum*. The last section of the right ureter is marked with a dotted line. Specimen is fixed in 76° alcohol. Right structures are represented by an empty pattern and left structures with a filled pattern: (**△**, ▲) bodies of the uteri; (**☆**, ★) *cervices*; (**➼**) *papillae uteri*; (**S**) *septum*; (**ф**) vaginal sinus; (**◇**, ◆) lateral *vaginae* with cranial loop (cr) and caudal loop (cd); (**“**) ureter. Red inset: inner dorsal mucosa of the urinary bladder showing the (**]**) paired ureteral openings (*ostium ureteris*), close to the neck, and the curved arrow pointing to the internal urethral orifice (*ostium urethrae internum*).

**Figure 6 animals-13-02377-f006:**
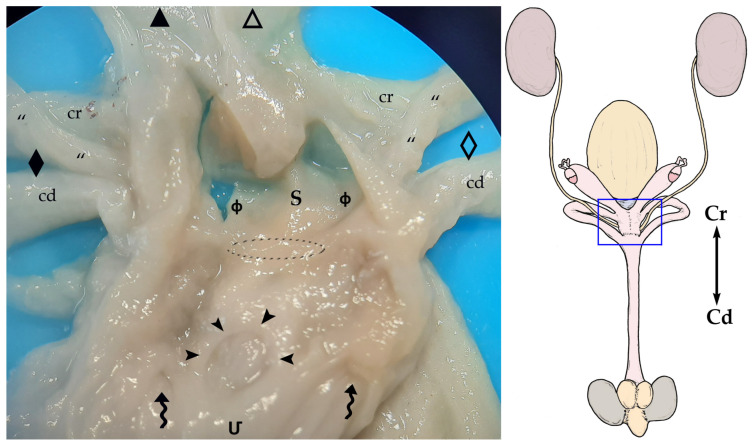
On the right, a general sketch of the female sugar glider urogenital system ex situ (dorsal view) with the area’s detailed photograph. On the left, view of the floor of the cranial part of the (**Մ**) urogenital sinus that has been opened by cutting its roof in the midline and moving each half laterally. The vaginal sinus (**ф**) has also been opened dorsally. In this specimen, the central vagina (marked with a discontinuous oval shape) connects the vaginal and urogenital sinuses. Specimen is fixed in formalin. (⇝) Outlet orifices of the lateral *vaginae*; (**➤**) external urethral orifice; (**“**) ureters; (**S**) *septum*. Right structures are represented by an empty pattern and left structures with a filled pattern: (**△**, ▲) bodies of the *uteri*; (**◇**, ◆) lateral *vaginae* with cranial loop (cr) and caudal loop (cd).

**Figure 7 animals-13-02377-f007:**
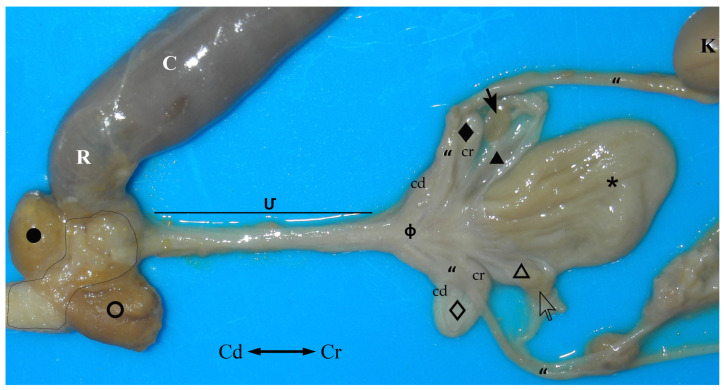
Dorsal view of the female sugar glider’s *apparatus genitalis* ex situ. The large intestine (colon and rectum) has been moved laterally to observe the urogenital sinus. Specimen is fixed in 76° alcohol. (**K**) Left kidney; (**“**) ureters; (_🞳_) urinary bladder. Right structures are represented by an empty pattern and left structures with a filled pattern: (**⇨**, **➞**) ovaries; (**△**, ▲) bodies of the uteri; (**◇**, ◆) lateral *vaginae* with cranial loop (cr) and caudal loop (cd); (**ф**) vaginal sinus; (**Մ**) *sinus urogenitalis*; (**○**, **●**) paracloacal glands; (**C**) colon; (**R**) rectum; (black line) cloaca.

**Figure 8 animals-13-02377-f008:**
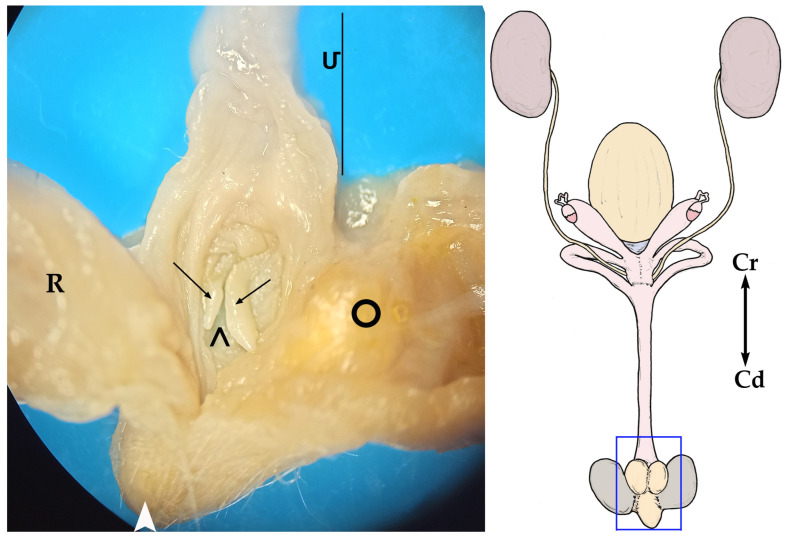
On the right, a general sketch of the female sugar glider urogenital system ex situ, (dorsal view) with the area’s detailed photograph. On the left, magnified image of the cloacal area; view of the floor of the caudal part of the urogenital sinus (**Մ**). The sinus has been opened by cutting its roof in the midline and moving each half laterally. (**R**) Rectum; (
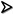
) cloacal orifice; (➝) clitoris; (**>**) *fossa clitoridis*; (**○**) right paracloacal gland.

## Data Availability

Data sharing is not applicable to this article as all data associated is available in the text.

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
