# Peer review of "Gross Anatomy of the Female Reproductive System of Sugar Gliders (Petaurus breviceps)"

_animals, 2023, doi:10.3390/ani13142377_

Round 1
Reviewer 1 Report
Line 53, Taxonomy of P. breviceps and its distribution may be more complex.
Indonesia as most people understand it is on the other side of the "Wallace line"
and the "Webber line" and P. breviceps are not naturally found there.
Line 55, "Nocturnal" rather than nocturne and "omnivorous" rather than omnivore
line 69, marsupium rather than marsupio
line 112, Figure 3, For the ovaries, the word precedes the symbol, but for everything else the symbol proceeds the words
line 124, use mesentery rather than mesos
line 128, word precedes symbol and on following lines word follows symbol
line 129, use mesentery rather than mesos
Line 129, empty white arrow is harder to see (especially when printed in black and white) and requires description,
maybe use a different symbol
Line 285 Bursa rather that bourse
390, Castrate = "the act of removing the testicles of a male animal or person" ref Oxford dictionary.
Better to use "ovariohysterectomy" or hysterectomy depending on actual technique proposed.
Line 392, If planning surgery, a clinician would also need to understand the relationship of the ureters
to the female reproductive tract. This is possibly of greater importance due to the consequences
of including the ureters in ligatures, and would be worth a mention somewhere in the article.
Line 55, "Nocturnal" rather than nocturne and "omnivorous" rather than omnivore
line 69, marsupium rather than marsupio
line 112, Figure 3, For the ovaries, the word precedes the symbol, but for everything else the symbol proceeds the words
line 124, use mesentery rather than mesos
line 128, word precedes symbol and on following lines word follows symbol
line 129, use mesentery rather than mesos
Line 129, empty white arrow is harder to see (especially when printed in black and white) and requires description,
maybe use a different symbol
Line 285 Bursa rather that bourse
390, Castrate = "the act of removing the testicles of a male animal or person" ref Oxford dictionary.
Better to use "ovariohysterectomy" or hysterectomy depending on actual technique proposed.
Author Response
10th July, 2023
Manuscript Animals-2482474: Yllera et al.
Reviewer #1:
Dear Reviewer:
We appreciate all your comments and suggestions. We respond to each point below, making changes to the manuscript accordingly.
à Line 53, Taxonomy of P. breviceps and its distribution may be more complex. Indonesia as most people understand it is on the other side of the "Wallace line" and the "Weber line" and P. breviceps are not naturally found there.
The misleading terms "Indonesia" and "Papua" have been deleted. New Guinea is the second-largest island in the world. The eastern half of the island is the largest land mass of the independent state of Papua New Guinea. The western half, known as Western New Guinea, is part of Indonesia [1].
According to the genetic study carried out by Campbell et al. (2019) [2], the USA sugar gliders have originated from a source population in the vicinity of Sorong, Western New Guinea, Indonesia.
à Line 55, "Nocturnal" rather than nocturne and "omnivorous" rather than omnivore.
The terminology has been changed.
à Line 69, marsupium rather than marsupio.
The changes have been made (L69 and L78).
à Line 112, Figure 3, For the ovaries, the word precedes the symbol, but for everything else the symbol proceeds the words.
Now, the symbol comes before the structure name. Also applied to L108 with the urinary bladder symbol.
à Line 124, use mesentery rather than mesos.
According to the Webster's Medical Dictionary [Chapman and Hall Medical (1986), Ed. Merrian-Webster Inc.] Mesentery: 1: ‘One or more vertebrate membranes that consist of a double fold of the peritoneum and invest the intestines and their appendages and connect them to the dorsal wall of the abdominal cavity; specif: such membranes connected with the jejunum and ileum in man’. 2: ‘a fold of membrane comparable to a mesentery and supporting a viscus (as the heart) that is not part of the digestive tract’. However, in the Nomina Anatomica Veterinaria (NAV) [Illustrated Veterinary Anatomical Nomenclature (2007) Edited by Oscar Schaller. 2nd ed. Ed. Enke Verlag], Mesentery (p228) is referred to the peritoneal folds associated to the intestine. In fact, ’Mesenterium’ is the mesentery in the specific sense, attached to Jejunum and Ileum; and all parts of the intestine (small and large intestine) is suspended by a mesentery, with different names depending on the part. Hence, as the NAV says that mesentery is referred to peritoneum and no other serous membrane, the second meaning in Webster’s Medical Dictionary is not applicable due to no viscus from the thoracic cavity is supported by a mesentery. Taking this into account, and strictly following veterinary terminology, instead of ‘mesos’ or ‘mesentery’ (as you suggest), we would prefer calling it ‘peritoneal folds’. This term was included in L128, L133 and L137.
à Line 128, word precedes symbol and on following lines word follows symbol.
Now, the symbol goes before the structure name.
à Line 129, use mesentery rather than mesos.
As justified for the same case in L128, ‘mesos’ has been changed to ‘peritoneal fold’.
à Line 129, empty white arrow is harder to see (especially when printed in black and white) and requires description, maybe use a different symbol.
We agree that the empty white arrow stands out poorly in the Figure 4. So, the symbol has been changed to a more visible one (!).
à Line 285 Bursa rather that bourse.
The change has been made.
à Line 390, Castrate = "the act of removing the testicles of a male animal or person" ref Oxford dictionary.
Better to use "ovariohysterectomy" or hysterectomy depending on actual technique proposed.
It may be that people in general use ‘castrate’ only for males. However, in veterinary and medical nomenclature, as reflected in Webster's Medical Dictionary [Chapman and Hall Medical (1986), Ed. Merrian-Webster Inc.] the meaning of 1castrate: vt, 1a: to deprive of the testes: Geld; 1b: to deprive of the ovaries: Spay. 2castrate: n, a castrated individual. However, we have changed ‘castrate a female’ to ‘perform an ovariohysterectomy’ (L409).
à Line 392, If planning surgery, a clinician would also need to understand the relationship of the ureters to the female reproductive tract. This is possibly of greater importance due to the consequences of including the ureters in ligatures and would be worth a mention somewhere in the article.
The course of the ureters is described in L192-195. However, a more detailed description (L203-208) and a photograph of the openings of the ureters at the roof of the urinary bladder have been introduced (Figure 5, red inset). In addition, in the discussion chapter, it is mentioned that surgeons must be very careful with the course of the ureters so as not to compromise their physical integrity during surgical interventions such as ovariohysterectomy or, especially, with ovario-vaginal-hysterectomy due to their close topographical relationship (L360-372).
àComments on the Quality of English Language
Line 55, "Nocturnal" rather than nocturne and "omnivorous" rather than omnivore
line 69, marsupium rather than marsupio
line 112, Figure 3, For the ovaries, the word precedes the symbol, but for everything else the symbol proceeds the words
line 124, use mesentery rather than mesos
line 128, word precedes symbol and on following lines word follows symbol
line 129, use mesentery rather than mesos
Line 129, empty white arrow is harder to see (especially when printed in black and white) and requires description,
maybe use a different symbol
Line 285 Bursa rather that bourse
390, Castrate = "the act of removing the testicles of a male animal or person" ref Oxford dictionary.
Better to use "ovariohysterectomy" or hysterectomy depending on actual technique proposed.
All these items are a repetition of the previous ones, so they have already been answered.
Hoping these modifications have improved our manuscript and you now consider it suitable for publication in this Special Issue of Animals.
Best regards,
Matilde Lombardero

Reviewer 2 Report
The paper is very good. Everything is very well explained, very clar. On top of that, the species petaurus is of great interest and a very good contribution for anatomists. The only thing i suggest, may be foor future publications, is to use numbers to point out structures in the photos. Symbols are sometimes difficult to understand. This is my only recommendation.
Author Response
10th July, 2023
Manuscript Animals-2482474: Yllera et al.
Reviewer #2:
The paper is very good. Everything is very well explained, very clear. On top of that, the species petaurus is of great interest and a very good contribution for anatomists. The only thing I suggest, may be foor future publications, is to use numbers to point out structures in the photos. Symbols are sometimes difficult to understand. This is my only recommendation.
Dear Reviewer,
We are very grateful for your positive comments.
Regarding your suggestion of using numbers instead of symbols, we would like to point out that using numbers to mark the structures was our decision at the beginning of this work. However, as we planned to always assign the same number for each structure, the list of non-correlative numbers in the figure legends might lead to confusion afterwards. Therefore, we decided to use symbols (always the same for the same structure), varying the right side with respect to the left side (presenting the outline as the right side and the filled symbol for the left side). Nevertheless, we will try to take your suggestion into account in future works.
Best regards,
Matilde Lombardero

Reviewer 3 Report
Gross Anatomy of the Female Reproductive System of Sugar Gliders (Petaurus breviceps)
This study describes the morphology of the female genital organs in the Sugar Gliders (Petaurus breviceps), which is of interest to the veterinary practice in the exotics clinics. This study has important scientific content, and I encourage the results to be published. However, in my opinion, it is not yet the time for publication. I am going to explain the basis of my opinion: a) The number of females is small, only 7 individuals, and b) genital organs show multiple changes depending on the sexual state (luteal, follicular, pregnant) of the female. The main objective of these studies consists of understanding morphophysiology, which has multiple clinical applications. I believe that the level of the journal requires a complete study of the morphology of the female genital organs including all sexual, or at least non-pregnant in the follicular and luteal phases, and also a complementary histological evaluation better explaining the observed changes. This histology could be very useful to explain structures that are not very common in other species.
I want to emphasize that my decision does not mean that the study at present is wrong. My main focus is it is not enough.
The graphical material in these studies is very important. Comments:
Figure 1: Magnificent. Figure 1c need a zoom.
Figure 2: Pretty. I suggest an ex situ image of the genital organs
Figure 3: Magnificent.
Figure 4: Correct, although Figure 4a image has a confusing position. It is preferable to take dorsal and ventral views, and avoid oblique views.
Figure 5, 6, 7, 8 and 9: Too many images, and lack of contextualization. Tips: a) Place a global image of the organs and a square indicating the part explained, and b) organ for 2-3 hours in formalin may improve visualization of structures .
Author Response
10th July, 2023
Manuscript Animals-2482474: Yllera et al.
Reviewer #3:
Reviewer #3:
Dear Reviewer,
We are grateful for your comments and suggestions. Below we try to respond to each point raised:
à This study describes the morphology of the female genital organs in the Sugar Gliders (Petaurus breviceps), which is of interest to the veterinary practice in the exotics clinics. This study has important scientific content, and I encourage the results to be published. However, in my opinion, it is not yet the time for publication. I am going to explain the basis of my opinion: a) The number of females is small, only 7 individuals,
We have increased the number of specimens to eight (L75-80). Before it died, this female had recently weaned two joeys. We have carried out a systematic dissection in which we have not visualized any alteration in its morphology with respect to the previous description made in this study. Hence, this reinforces our results. (We wanted to comment that we have not incorporated photographs in the manuscript from this new specimen because she had cadaveric hypostasis in the lumbar area).
à and b) genital organs show multiple changes depending on the sexual state (luteal, follicular, pregnant) of the female. The main objective of these studies consists of understanding morphophysiology, which has multiple clinical applications. I believe that the level of the journal requires a complete study of the morphology of the female genital organs including all sexual, or at least non-pregnant in the follicular and luteal phases, and also a complementary histological evaluation better explaining the observed changes. This histology could be very useful to explain structures that are not very common in other species.
We are veterinary anatomists and our idea is to study and describe the macroscopic/gross anatomy of the female urogenital apparatus in order to provide a useful tool for clinicians who must perform surgery and correctly interpret the images obtained by the diagnostic imaging devices available today, in order to make a correct diagnosis of the pathologies of their patients. This work is a first approach to the study of the gross anatomy of the female sugar glider reproductive system. Nevertheless, we take note of the idea to carry out a further detailed histological study of the female genital apparatus in different phases of their reproductive cycle. This would require numerous animals with their reproductive cycles synchronized and being sacrificed sequentially to cover all the phases of their reproductive cycle. This would be an ambitious project for which, at the moment, we do not have funding.
à I want to emphasize that my decision does not mean that the study at present is wrong. My main focus is it is not enough.
This work is a first approach to the study of the gross anatomy of the female sugar glider reproductive system. However, it has a direct application to identify the structures on diagnostic images, which are essential to reach a correct diagnosis in order to treat the patient properly.
à The graphical material in these studies is very important. Comments:
Figure 1: Magnificent. Figure 1c need a zoom.
Accordingly with your comment, the figure 1c has been zoomed, and letters (a,b,c) have been changed to black. We hope you approve the changes made.
à Figure 2: Pretty. I suggest an ex situ image of the genital organs
Precisely, to provide more visual information on the arrangement of the urogenital apparatus of the female sugar glider, in some images (Figures 5, 6 and 8) we have included an ex situ sketch (similar to Figure 7) to indicate the area that has been enlarged in the photographs.
à Figure 3: Magnificent.
à Figure 4: Correct, although Figure 4a image has a confusing position. It is preferable to take dorsal and ventral views, and avoid oblique views.
In a ventral view, the suspensory ligament of the ovary is not visible as it is located dorsally to the ovary (curved arrow in accompanying image). Therefore, we believe that an oblique view is the only way to properly visualize the ligament. This is the image taken ventrally, in which it is impossible to show where the suspensory ligament of the ovary is located.
Consequently, we consider that figure 4a should be kept as it is, although we have modified its angle slightly (15º), so that the abduced thighs are horizontal, rather than oblique.
à Figure 5, 6, 7, 8 and 9: Too many images, and lack of contextualization. Tips: a) Place a global image of the organs and a square indicating the part explained, and b) organ for 2-3 hours in formalin may improve visualization of structures.
According to your suggestion, we decided to remove the former Figure 6 and replace it with the former figure 8, but adding more information. All figures are now oriented in the same way, and in some of them, a diagram of the female urogenital apparatus has been included to show the area depicted in the photographs.
We hope you consider that the manuscript has improved and is now suitable for publication in Animals.
Best regards,
Matilde Lombardero

Round 2
Reviewer 3 Report
Gross Anatomy of the Female Reproductive System of Sugar Gliders (Petaurus breviceps)
Thank you very much for the new version of the article. I understand that the article is mainly addressed to surgeons; however, I still think about the need to carry out a microscopic study of the morphology of the female genital organs, always looking for multidisciplinary (that is, including histology) and considering its changing morphophysiology. In this way, I encourage the authors to consider it in future works. Once the macroscopic study is focused, it is true that such a large sample size is no longer necessary, and the main anatomical characteristics can be described based on eight individuals.
For all document, I suggest the change of the “female genital apparatus” per “female genital organs”. Authors are using different terminology, that should be standardized: “female genitalia”, “genital organs”, “genital apparatus”, “genital system”, “reproductive trat”, “reproductive organs”, “reproductive system”, … I suggest using “genital organs” in the whole manuscript.
Simple Summary:
L 20. “Sugar gliders have duplicates of all the genital organs: they have two ovaries, two oviducts, two uteri and two vaginas. In addition, in pregnant females a new conduit arises: the birth canal. In addition, in pregnant females a new conduit arises: the birth canal, through which the young will leave the genital tract to go outside”. I suggest: “Sugar gliders have duplicates of all the genital organs: they have two ovaries, two oviducts, two uteri and two vaginas, which opens independently to a common cloaca”
L. 23. Delete: “Furthermore,”
Abstract:
L. 27. “We dissected carcasses of eight mature females, both parous and non-parous specimens, to study the macroscopic anatomy of the female genital organs in the sugar glider”
The events involving the median vagina, also called the 'birth canal' in pregnant females is amazing. According to the authors’ observations, this channel is formed exclusively in pregnant females, and disappears in non-pregnant females. I believe that the duct always exists and that it closes in non-pregnant females, and opens in pregnant females; but the structure exists in all females. In fact, as the authors report in the Discusion section (L 340), “However, in most Macropodidae (kangaroos), the median vagina becomes a permanent structure after the first birth [12]. Judah and Nutall (2008) [8] point out that it is a temporary structure in Petaurus breviceps, which coincides with our own observations.”. Thus, this assumption suggests that the birth canal is always present, but closed in non-pregnant females. I suggest this change when needed accordingly.
Introduction:
L 46. “Very often, it is hard to find appropriate references in the literature and even sometimes, the available information is contradictory, and erroneous extrapolations from domestic species are applied.”
L 63. “… but they differ in their means of reproduction,”… I suggest using the term “reproductive performance”
L 64. “genital apparatus” per “genital organs”.
L 66. “… highly altricial young emerges through the cloacal opening and climb on its own into a pouch called a marsupium, where its development will be completed.”
L 76 “carcasses”
L 75. “Seven animals had given birth to joeys at least once, including three females with joeys in the pouch, one with two joeys recently weaned, and three individuals that had not been pregnant for some years; one additional female was no parous.”
Results
L 121. “In this study, all the females were pubescent, so the gonad’s surface was irregular due to the projection of follicles and/or corpora lutea.”. Thus, all functional ovarian structures are visible.
L 123-28. Could the clitoris be considered an external genital organ?
Discusion
L 250. The morphology of the pouch in the sugar glider matches the description made by 250 Smith in 1973 [7].
L 281. “Thus, the ovary produces gametes and reproductive hormones, fertilization takes place in the oviduct (pair) and the uterus (unpaired) provides the environment and adequate nourishment for the embryo development until its birth.”. Be careful, there are eutherians that have a double uterus, such as histricognaths rodents (Cuniculus paca or Dasyprocta sp). It is likely that the latter are in an intermediate evolutionary stage with the marsupials.
Mayor, P., Guimaraes, D., López-Plana, C. 2013. Functional morphology of the genital organs in the wild paca (Cuniculus paca) female. Animal Reproduction Science, 140(3-4):206-215. doi: 10.1016/j.anireprosci.2013.06.010.
Mayor, P., Bodmer, R. and López-Béjar, M. 2011. Function anatomy of the female genital organs of the black Agouti (Dasyprocta fuliginosa) in the Peruvian Amazon. Anim Reprod Sc.
123: 249-257. Doi 10.1016/j.anireprosci.2010.12.006
L 364. placental or eutherian mammals? My comment is aimed at maintaining constant terminology.
L 377. “As stated by Pearson (1945) [31], the long urogenital sinus of the sugar glider seems to be characteristic of marsupials.”. This sentence seems disconnected.
Conclusion
L 410. “Although dystocias are not frequent because the joeys are very small when they are born, as mammals they can suffer from uterine inertia, malpositioned foetus, malformation of the reproductive tract and so on.” This sentence requires some references.
I suggest the inclusion of some final statement similar to: “Our results provide information useful for clinical and surgical practice, and also to improve the understanding of the evolutionary processes that characterize eutherians and marsupial mammals.
Figures
Change “Magnified” per “detailed”.
Author Response
18th July, 2023
Gross Anatomy of the Female Reproductive System of Sugar Gliders (Petaurus breviceps)-R2
Dear Reviewer,
These are the responses to your comments:
•Thank you very much for the new version of the article. I understand that the article is mainly addressed to surgeons; however, I still think about the need to carry out a microscopic study of the morphology of the female genital organs, always looking for multidisciplinary (that is, including histology) and considering its changing morphophysiology. In this way, I encourage the authors to consider it in future works. Once the macroscopic study is focused, it is true that such a large sample size is no longer necessary, and the main anatomical characteristics can be described based on eight individuals.
We do appreciate your constructive comments. We thank you for your suggestion and we will take it into account in future studies.
All the changes made to the manuscript are highlighted in yellow in the tracked version.
• For all document, I suggest the change of the “female genital apparatus” per “female genital organs”. Authors are using different terminology, that should be standardized: “female genitalia”, “genital organs”, “genital apparatus”, “genital system”, “reproductive trat”, “reproductive organs”, “reproductive system”, … I suggest using “genital organs” in the whole manuscript.
We use this variety of terminology (genital/reproductive apparatus/system…) because all terms are synonymous. We believe that using extensive vocabulary and terminology enriches the text and does not lead to confusion, as it is terminology commonly used in Veterinary sciences. On the contrary, replacing all these expressions by ‘genital organs’ we think restricts the vocabulary quite a lot.
In addition, according to the Webster's Medical Dictionary [Chapman and Hall Medical (1986), Ed. Merrian-Webster Inc.]:
-Organ: a differentiated structure (as a heart or kidney) consisting of cells and tissues and performing some specific functions in an organism.
-Apparatus: a group of bodily parts and especially organs having a common function [the respiratory apparatus].
-System: 1a: a group of body organs that together perform one or more vital function ̶ see circulatory system, nervous system, reproductive system, respiratory system. 1b: The body considered as a functional unit.
Thus, by using the term ‘genital organs’, the idea of the integral functionality of the ‘system’ –which is a higher level of structural organization than the organs– , is lost.
Simple Summary:
• L 20. “Sugar gliders have duplicates of all the genital organs: they have two ovaries, two oviducts, two uteri and two vaginas. In addition, in pregnant females a new conduit arises: the birth canal. In addition, in pregnant females a new conduit arises: the birth canal, through which the young will leave the genital tract to go outside”. I suggest: “Sugar gliders have duplicates of all the genital organs: they have two ovaries, two oviducts, two uteri and two vaginas, which opens independently to a common cloaca.”
We appreciate your suggestion for the drafting of this paragraph, but there is an inaccuracy: the vaginas do not flow independently into the cloaca, but into the urogenital sinus, which also receives the urine stored in the bladder. It is the urogenital sinus that transports the urine and the foetus into the cloaca. Therefore, we do think that this paragraph should remain as it stands.
• L 23. Delete: ‘Furthermore,’
‘Furthermore’ has been deleted.
Abstract:
• L 27. “We dissected carcasses of eight mature females, both parous and non-parous specimens, to study the macroscopic anatomy of the female genital organs in the sugar glider”
‘Corpses’ has been changed to ‘carcasses’.
‘Seven pubertal females’ has been changed to ‘eight mature females’.
• The events involving the median vagina, also called the 'birth canal' in pregnant females is amazing. According to the authors’ observations, this channel is formed exclusively in pregnant females, and disappears in non-pregnant females. I believe that the duct always exists and that it closes in non-pregnant females, and opens in pregnant females; but the structure exists in all females. In fact, as the authors report in the Discusion section (L 340), “However, in most Macropodidae (kangaroos), the median vagina becomes a permanent structure after the first birth [12]. Judah and Nutall (2008) [8] point out that it is a temporary structure in Petaurus breviceps, which coincides with our own observations.”. Thus, this assumption suggests that the birth canal is always present, but closed in non-pregnant females. I suggest this change when needed accordingly.
Most of the publications found in the literature are macroscopic studies. They only show if the communication between the vaginal and urogenital sinuses is maintained. To confirm if the canal persists (even if its end is closed), it is necessary to study it in histological sections. To clarify the paragraph concerning the middle vagina in the discussion (L349-352 tracked version; L343- 345 clean version), the following text has been added:
‘However, the birth canal could persist even if it is blocked by tissue separating the vaginal and urogenital sinuses. Histological study would be required to ensure that it disappears completely after parturition, as it seems to be the case of the Tasmanian tiger [28]’.
Introduction:
• L 46. “Very often, it is hard to find appropriate references in the literature and even sometimes, the available information is contradictory, and erroneous extrapolations from domestic species are applied.”
The change has been made.
• L 63. “… but they differ in their means of reproduction,”… I suggest using the term “reproductive performance”
The term has been changed to the one you have suggested.
• L 64. “genital apparatus” per “genital organs”.
This issue has been discussed previously.
• L 66. “… highly altricial young emerges through the cloacal opening and climb on its own into a pouch called a marsupium, where its development will be completed.”
This information has been added to the text.
• L 76. “carcasses”
‘Corpses’ has been changed to ‘carcasses’.
• L 75. “Seven animals had given birth to joeys at least once, including three females with joeys in the pouch, one with two joeys recently weaned, and three individuals that had not been pregnant for some years; one additional female was no parous.”
The change has been made.
Results
• L 121. “In this study, all the females were pubescent, so the gonad’s surface was irregular due to the projection of follicles and/or corpora lutea”. Thus, all functional ovarian structures are visible.
The change has been made.
• L 123-28. Could the clitoris be considered an external genital organ?
We are not sure which lines you are referring to: maybe L216-219?
The Illustrated Veterinary Anatomical Nomenclature (Schaller, O, Ferdinand Enke Verlag Stuttgart, 1992) includes the clitoris in the external female genitalia (p218). Thus, in order to avoid misunderstanding, the sentence ‘the sugar glider has no external genitalia’ has been changed to ‘the female sugar glider has neither a vulva nor an anus’ (L216, tracked version; L212 clean version).
Discusion
• L 250. The morphology of the pouch in the sugar glider matches the description made by 250 Smith in 1973 [7].
As requested, the information has been included (L255 tracked version; L250 clean version).
• L 281. “Thus, the ovary produces gametes and reproductive hormones, fertilization takes place in the oviduct (pair) and the uterus (unpaired) provides the environment and adequate nourishment for the embryo development until its birth.”. Be careful, there are eutherians that have a double uterus, such as histricognaths rodents (Cuniculus paca or Dasyprocta sp). It is likely that the latter are in an intermediate evolutionary stage with the marsupials.
Mayor, P., Guimaraes, D., López-Plana, C. 2013. Functional morphology of the genital organs in the wild paca (Cuniculus paca) female. Animal Reproduction Science, 140(3-4):206-215. doi: 10.1016/j.anireprosci.2013.06.010.
Mayor, P., Bodmer, R. and López-Béjar, M. 2011. Function anatomy of the female genital organs of the black Agouti (Dasyprocta fuliginosa) in the Peruvian Amazon. Anim Reprod Sc. 123: 249-257. Doi 10.1016/j.anireprosci.2010.12.006
As you know, the uterus of eutherians consists of two horns, a body and a cervix leading to the vagina. However, lagomorphs and most rodents have a ‘duplex/double uterus’. In them, this organ consists of two horns (without a body) that independently lead to a single vagina. Therefore, they are the exception to the general rule of the odd uterus in eutherian mammals. However, in these species, the final part of both horns usually shares the external muscular layer, so there is a partial external fusion which looks like a body. This information is supported by:
- [16] Girling, S.J. 2013. Veterinary Nursing of Exotic Pets, 2nd ed. Wiley-Blackwell, USA.
- Dyce, K.M., Sack, W.O. and Wensing, C.J.G.2009. Textbook of Veterinary Anatomy, 4th ed, Saunders.
- Vella, D. and Donnelly, T.M. Basic Anatomy, Physiology, and Husbandry. In Ferrets, Rabbits, and Rodents, p157–173. Elsevier, 2012. doi:10.1016/B978-1-4160-6621-7.00012-9
- Jarrett CL, Jarrett TR, Harvey SB, Alworth L. The Uterus Duplex Bicollis, Vagina Simplex of Female Chinchillas. J Am Assoc Lab Anim Sci. 2016 55:155-160. PMID: 27025806; PMCID: PMC4783633.
among others. Thus, “unpaired, except in lagomorphs and most rodents which have duplex uteri” has been added (L287-288 tracked version; L282-283 clean version).
• L 364. placental or eutherian mammals? My comment is aimed at maintaining constant terminology.
The change has been made.
• L 377. “As stated by Pearson (1945) [31], the long urogenital sinus of the sugar glider seems to be characteristic of marsupials.” This sentence seems disconnected.
The sentence has been incorporated into the following paragraph.
Conclusion
• L 410. “Although dystocias are not frequent because the joeys are very small when they are born, as mammals they can suffer from uterine inertia, malpositioned foetus, malformation of the reproductive tract and so on.” This sentence requires some references.
As a general rule, in the conclusions of an article, references are not included, as they are the conclusions of the work itself, not a discussion. ‘Genital neoplasias’ has been included in the list of potential pathologies.
• I suggest the inclusion of some final statement similar to: “Our results provide information useful for clinical and surgical practice, and also to improve the understanding of the evolutionary processes that characterize eutherians and marsupial mammals.
According to your suggestions, we have included this paragraph as a final statement:
“Our results provide useful information for clinicians, especially for the correct interpretation of modern imaging techniques, such as magnetic resonance imaging (MRI) and computed tomography (CT), as well as to perform successful surgery in the female genital tract, when needed.
On the other hand, we think it would be better no to incorporate the final part of your sentence, because it was not in our objectives to compare the evolutionary processes of marsupial mammals with eutherian mammals, but their anatomy.
Figures
• Change “Magnified” per “detailed”.
The word has been changed in the three figures (5, 6 and 8).
We do hope that with these changes, our manuscript meets with your approval.
Best regards,
Matilde Lombardero